# Melatonin-Induced Inhibition of *Shiraia* Hypocrellin A Biosynthesis Is Mediated by Hydrogen Peroxide and Nitric Oxide

**DOI:** 10.3390/jof8080836

**Published:** 2022-08-10

**Authors:** Wenjuan Wang, Qunyan Huang, Yue Wang, Xinping Li, Jianwen Wang, Liping Zheng

**Affiliations:** 1College of Pharmaceutical Sciences, Soochow University, Suzhou 215123, China; 2Department of Horticultural Sciences, Soochow University, Suzhou 215123, China

**Keywords:** melatonin, *Shiraia*, hypocrellin, reactive oxygen species, nitric oxide

## Abstract

Melatonin (MLT), an evolutionarily conserved pleiotropic molecule, is implicated in numerous physiological processes in plants and animals. However, the effects of MLT on microbes have seldom been reported. In this study, we examined the influence of exogenous MLT on the growth and hypocrellin biosynthesis of bambusicolous fungus *Shiraia* sp. S9. Hypocrellin A (HA) is a photoactivated and photoinduced perylenequinone (PQ) toxin in *Shiraia*. Exogenous MLT at 100.00 μM not only decreased fungal conidiation and spore germination but inhibited HA contents significantly in fungal cultures under a light/dark (24 h:24 h) shift. MLT treatment was associated with higher activity of antioxidant enzymes (superoxide dismutase, catalase and peroxidase) and a marked decline in reactive oxygen species (ROS) production in the mycelia. Moreover, MLT induced endogenous nitric oxide (NO) production during the culture. The NO donor sodium nitroprusside (SNP) potentiated MLT-induced inhibition of O_2_^−^ production, but NO scavenger 2-(4-carboxyphenyl)-4,4,5,5-tetramethylimidazoline-1-oxyl-3-oxide (cPTIO) enhanced O_2_^−^ production, whereas MLT-induced NO level was increased by the ROS scavenger vitamin C (Vc). The changes in NO and H_2_O_2_ were proved to be involved in the MLT-induced downregulation of the expressions of HA biosynthetic genes, leading to the suppression of HA production. This study provides new insight into the regulatory roles of MLT on fungal secondary metabolism activities and a basis for understanding self-resistance in phototoxin-producing fungi.

## 1. Introduction

Melatonin (*N*-acetyl-5-methoxytryptamine, MLT), a tryptophan derivative, was originally identified as an animal hormone produced by the bovine pineal gland in 1958, which has been associated with many physiological processes such as the regulation of circadian rhythms, immunological functions and antioxidant actions [1,2]. MLT was also found in many invertebrates, algae and plants [3]. MLT plays important roles in the regulation of plant growth and development, including stimulating seed germination, promoting photosynthesis and root growth, and enhancing stress tolerance [4,5]. MLT exhibits plant protection against multiple abiotic stresses such as salinity stress [6], water stress [7] and osmotic stress [8] and pathogen infection [9]. The protective effects are attributed to its free radical scavenging and antioxidant capacity to alleviate oxidative damage mediated by reactive oxygen species (ROS), including superoxide anion, hydrogen peroxide, and hydroxyl radicals [10]. Nitric oxide (NO) is a key signaling molecule that plays a crucial role in plant defense responses mediated or caused by ROS under stresses [11]. NO could control ROS production by altering the activities of several ROS-scavenging enzymes and act as an antioxidant to improve plant stress tolerance, whereas over-accumulation of NO may cause nitrosative stress leading to toxic damage to plant cells [12]. A lower concentration of local NO is essentially required to impart resistance in plants against different stresses. Recently, Shi et al. (2015) demonstrated that nitric oxide (NO) is a downstream signal of MLT in *Arabidopsis* against the bacterial pathogen *Pseudomonas syringe* infection [13]. NO-dependent melatonin synthesis was also observed in NaCl-stressed rapeseed seedling roots [14] and cadmium-stressed rice seedlings [15]. The counteracting effects of MLT on oxidative damages in those stresses were confirmed to be mediated by NO generation at lower concentrations.

Compared with many studies of MLT in animals and plants, the research on MLT in microbes is scarce. MLT was first detected from the unicellular organism *Gonyaulax polyedra* by Pöggeler et al. (1991) [16] and a photosynthetic bacterium *Rhodospirillum rubrum* by Manchester et al. (1995) [17]. These studies extended MLT research from plants and animals to microorganisms. Higher MLT content (9748 ± 2430 pg/mg protein) was present in an aerobic photosynthetic bacterium *Erythrobacter longus* [18]. Exogenous MLT at 10^−4^ M could alter the circadian rhythm of NAD^+^ kinase in *Neurospora crassa* and bioluminescence in *Gonyaulax polyedra* [19]. Vázquez et al. (2017) also reported that MLT at 5 μM could mitigate H_2_O_2_-induced damage to yeast by decreasing ROS accumulation, enhancing GSSG/GSH, and increasing the expressions of endogenous antioxidant defense genes [20]. There were a few studies on MLT production of yeasts in fermented foods and beverages [21]. However, little information is available regarding the effects of MLT on fungal growth and metabolism.

*Shiraia bambusicola* is a fungal pathogen on bamboos, and its fruiting body has been traditionally used in Chinese medicine to treat rheumatoid arthritis, tracheitis, stomachache and psoriasis [22]. Hypocrellins (HA-HC) with a perylenequinone (PQ) skeleton are the main bioactive components of *Shiraia* hypha and fruiting bodies [23]. The 4,9-dihydroxy-3,10-perylenequinoid moiety in hypocrellins is the essential structure for generating singlet oxygen (^1^O_2_) efficiently in photooxidative action [24]. The induced ROS could cause plant cellular damage and nutrient leakage, allowing for successful pathogenesis [25]. Due to superior singlet oxygen quantum yield and lower dark toxicity, hypocrellins have also been developed as new non-porphyrin photosensitizers for photodynamic therapy (PDT) on cancers and viruses [26,27]. On the other hand, the production of ^1^O_2_ and other ROS in photosensitization reactions of hypocrellins could cause oxidative stress in the filamentous fungal *Shiraia* itself [28]. In hyphae of *Shiraia* sp. S8, we have recently detected the presence of MLT at 0.24–1.39 mg/g dry weight (DW) [29]. However, whether MLT could act as an antioxidant to improve fungal resistance to hypocrellin phototoxicity is still unknown. Therefore, as a follow-up exploration of signaling regulation of ROS, NO and extracellular ATP (eATP) on *Shiraia* hypocrellin biosynthesis in our lab [30,31], we wish to explore the physiological role of MLT on *Shiraia* growth and hypocrellin production. We investigated the possible mediation of ROS and NO during MLT action, especially on the inhibition of expressions of HA biosynthetic genes and HA production. This study may help us understand the resistance mechanisms by PQ-producing fungi and the signaling regulation of MLT on fungal PQ biosynthesis.

## 2. Materials and Methods

### 2.1. Strains and Culture Conditions

The fungal *Shiraia* sp. S9 registered in China General Microbiological Culture Collection Center as No. CGMCC16369 was used for hypocrellin production [32]. The seed culture preparation, liquid culture medium composition and culture conditions have been described in detail in our previous report [33]. Briefly, the seed culture (1 mL) was transferred to 150 mL Erlenmeyer flasks containing 50 mL liquid medium (100 g/L potato, 20 g/L starch, 4 g/L NaNO_3_, 1.5 g/L KH_2_PO_4_, 0.5 g/L CaCO_3_ and 0.01 g/L VB_1_, pH 6.3) and cultured at 150 rpm and 28 °C for 8 days. Both solid and liquid cultures of S9 were exposed under our optimized light condition for hypocrellin production: intermittent illumination (light/dark shift, 24 h:24 h) with an intensity of 200 lx during the cultures. The light-emitting diode lamps (0.9 m, 18 W/m^2^, Xiaoyecao Photoelectric Technology Co., Ltd., Shenzhen, China) were installed parallel to the lightbox (1 × 1 × 0.8 m) above the shaking table (ZD-8802, Hualida Co., Ltd., Suzhou, China). The light conditions were detailed in our previous study [34]. 

### 2.2. MLT Treatment

MLT (Solarbio Science & Technology, Beijing, China) was dissolved in absolute ethanol and diluted with deionized water, then filtered with a 0.22 μm organic phase filter (Titan Science, Shanghai, China). MLT at 0.05–500.00 μM was mixed with 25 mL potato dextrose agar (PDA) medium and poured into sterile 90-mm Petri dishes in solid culture. As to the shake-flask cultures, MLT (0.05–500.00 μM) was added to the liquid medium on day 2 of the culture.

### 2.3. Morphology Observation and Conidia Quantification

To observe the sporulation of S9, the conidia were collected in 10 mL distilled water on day 8 of cultures and quantified using a hemocytometer under a microscope (CX21, Olympus, Tokyo, Japan). In germination experiments, spores (1 × 10^7^/mL) were inoculated in a 3 cm-diameter Petri dish with 5 mL of liquid medium. After cultivation with MLT at 100.00 μM for 6–24 h, the germination rate was determined by counting 300 spores randomly for each Petri dish. The fungal pellets in liquid cultures were observed using a stereoscopic microscope (SMZ1000, Nikon, Tokyo, Japan) and photographed at different cultivation times (days 2–8). The diameters of fugal pellets in liquid culture were measured in triplicates (50 pellets per replicate). 

### 2.4. Detection of ROS and Activities of Antioxidant Enzymes 

The generation of ROS in hyphae was observed with the ROS-specific fluorescent probe 2, 7-dichlorodihydrofluorescein diacetate (DCFH-DA, Beyotime Biotechnology, Haimen, China) [35]. DCFH-DA at 10 µM was added to the mycelium cultures and then incubated in the dark for 30 min. The mycelium was washed three times with deionized water, and the fluorescence was observed under an Olympus fluorescent microscope (CKX41, Tokyo, Japan) with an excitation/emission wavelength of 485/528 nm. Hydrogen peroxide (H_2_O_2_) and superoxide anion (O_2_^−^) contents in mycelia were determined as previously described [36,37]. The activities of superoxide dismutase (SOD), catalase (CAT) and NADPH oxidase (NOX) were measured using the Enzyme Activity Assay Kit (Beyotime Biotechnology, Nanjing, China) following the manufacturer’s instructions and previous reports [38,39]. The activity of peroxidase (POD) was determined as previously described [40]. To analyze the role of ROS in MLT treatment, exogenous H_2_O_2_ (100 μM) and vitamin C (Vc) (10 μM) were used as an ROS donor and scavenger, respectively, and their dosages were chosen based on our previous studies [30]. They were added to the culture to pre-treat for 30 min before the addition of MLT. 

### 2.5. Detection of NO Generation

For the detection of NO generation in hyphae, 4,5-diaminofluorescein diacetate (DAF-2 DA, Sigma-Aldrich, St. Louis, MO, USA) was used [41]. After 2 days of culture, DAF-2 DA at 10 μΜ was added to the culture 30 min prior to MLT treatment. The fluorescence was observed under an Olympus fluorescent microscope (CKX41, Tokyo, Japan) with an excitation/emission wavelength of 470/525 nm. NO contents in mycelia were determined using the Nitric Oxide Assay Kit (Beyotime Biotechnology, Nanjing, China) [30]. To investigate on the role of NO in MLT treatment, sodium nitroprusside (SNP, 10 μΜ) and 2-(4-carboxyphenyl)-4,4,5,5-tetramethylimidazoline-1-oxyl-3-oxide (cPTIO, 100 μΜ) were used as a NO donor and scavenger, respectively. Nω-nitro-L-arginine methylester (L-NAME, 100 μΜ) and sodium tungstate dehydrate (STD, 100 μΜ) were used as inhibitors of nitric oxide synthase (NOS) and nitrate reductase (NR), respectively. These reagents were added 30 min prior to the addition of MLT, and their dosages were chosen based on our previous studies [30].

### 2.6. Extraction and Quantification of HA

After the incubation, the mycelia were harvested and washed five times with deionized water and freeze-dried. Additionally, fungal biomass was assessed based on the dry weight. HA was extracted and detected according to the method from our previous report [42]. The reverse-phase Agilent 1260 HPLC (Agilent, Wilmington, NC, USA) system equipped with a 250 × 4.6 mm Agilent HC-C18 column (Agilent, Santa Clara, CA, USA) was used for the quantification. The sample was filtered through a 0.45-µm organic phase filter and eluted with a mobile phase (acetonitrile:water = 65:35, *v*/*v*) with a flow rate of 1 mL/min and monitored at 465 nm.

### 2.7. Quantitative Real-Time PCR Analysis

The primer of target genes for HA biosynthesis and internal reference gene (18S ribosomal RNA) are listed in Appendix A. The relative gene expressions were measured using qRT-PCR on the basis of our previous report [43].

### 2.8. Statistical Analysis

Each group consisted of three independent experiments (ten flasks per replicate). Student’s t-test and one-way analysis of variance (ANOVA) with Dunnet’s multiple comparison tests were performed for the results. Experimental values were presented as mean ± standard deviation (SD), and a *p*-value < 0.05 was considered statistically significant.

## 3. Results

### 3.1. Effect of MLT on Fungal Growth and HA Accumulation

In the solid culture of *Shiraia* sp. S9 under light/dark (24 h:24 h) shift, dark-brown pigments were produced after 8 days (the control group in Figure 1A). MLT treatment at 0.05–500.00 μM inhibited the pigment accumulation and turned the color from dark brown to pale yellow. HA contents were decreased by 17.10–81.29% in a dose-dependent manner under MLT treatment (Figure 1B). The conidia count showed no significant changes at low concentrations (0.05–0.50 μM) of MLT but decreased significantly at high concentrations (50.00–500.00 μM) (Figure 1C). The germination rate of spores decreased by 3.20–76.93% during the MLT treatment at 100.00 μM after 8–12 h (Figure 1D).

In the liquid cultures, the morphology of the fungal pellets was observed after MLT treatment at 100.00 μM. The bigger and fluffier pale pellets were formed after MLT treatment (Figure 2A). The diameter of fungal pellets was increased by 10.91–18.23% from day 5 to day 8 (Figure 2B). To investigate MLT’s effect on HA production in the mycelium culture, MLT (0.05–500.00 μM) was added on day 2 of the culture (Figure 2C,D). Although the fungal biomass was not altered by MLT, HA contents in fungal mycelia were suppressed significantly with the increasing concentration of MLT. The lowest content (1.32 mg/g DW) wa reached under MLT at 500.00 μM, about 11.40% of the control (Figure 2D). These results demonstrated that MLT inhibited HA biosynthesis of *Shiraia* sp. S9 in either solid or liquid cultures.

### 3.2. Effects of MLT on ROS Generation and the Activities of Antioxidant Enzymes

To examine ROS accumulation in the mycelia of *Shiraia* sp. S9, a fluorescent probe DCFH-DA was used. We observed a green fluorescence in mycelia in the culture under the exposure for 24–72 h of light/dark shift (24 h:24 h) (the control group in Figure 3A), and the green fluorescence was attenuated after the addition of MLT at 100 μM (MLT group in Figure 3A). The fluorescence intensity decreased by 39.49–75.33% after 24–72 h of MLT treatment compared with the control (Figure 3B). After 48 h of MLT treatment, the contents of O_2_^−^ and H_2_O_2_ in mycelia were found to be decreased by 45.63% and 17.97%, respectively (Figure 3C,D). When the culture was added with exogenous H_2_O_2_ (100 μM) with MLT, the suppressed H_2_O_2_ contents in mycelia were rescued (Figure 3E). These results indicated that MLT treatment could inhibit ROS production in *Shiraia* sp. S9 under light exposure.

In addition, the activities of NOX, SOD, CAT, and POD were also determined under MLT treatment. Although NOX activity was not altered (Figure 4A), the activities of antioxidant enzymes (CAT, SOD and POD) were increased significantly during the culture (Figure 4B–D). The activity of CAT was enhanced immediately after MLT addition on day 2 to 24.54 U/mg protein on day 3, about 1.38-fold over the control (Figure 4B). The SOD activity showed an increasing tendency with the cultivation time (Figure 4C). After MLT addition on day 2, POD activity increased to a broad peak on day 3 (0.13 U/mg protein) and to a higher peak on day 6 (0.20 U/mg protein) (Figure 4D).

### 3.3. Effect of MLT on NO Generation

We detected NO generation in *Shiraia* sp. S9 by the fluorescent probe DAF-2 DA. *Shiraia* hyphae displayed an obvious green fluorescence enhancement in response to MLT treatment at 100 μM (Figure 5A). After 8 and 24 h of MLT treatment, the relative intensity of DAF-2 DA fluorescence increased by 3.72- and 1.70-fold over the control, respectively (Figure 5B). A significant increase in the NO contents in mycelia was observed on days 4–8, whereas the peak value (8.61 μmol/g fresh weight, FW) was reached on day 5, about 3.27-fold higher than the control (Figure 5C). However, MLT-induced NO production was suppressed by 19.04%, 68.51% and 57.52% by NOS inhibitor L-NAME, NR inhibitor STD and NO scavenger cPTIO, respectively (L-NAME, STD or cPTIO + MLT vs. MLT in Figure 5D). These results suggested that MLT treatment could induce NO production in *Shiraia* sp. S9. To further investigate the interrelation between NO and ROS during MLT treatment, NO donor SNP (10 μM) and scavenger cPTIO (100 μM), and exogenous H_2_O_2_ (100 μM) and Vc (100 μM) were added, respectively, to the culture under MLT treatment. Although the addition of H_2_O_2_ did not cause significant changes in NO production induced by MLT (MLT vs. MLT + H_2_O_2_ in Figure 6A), Vc increased NO production by 67.26% (MLT vs. MLT + Vc in Figure 6A). Contrarily, pretreatment with NO donor SNP could reduce O_2_^−^ contents by 61.2% (MLT vs. MLT + SNP in Figure 6B), but cPTIO increased O_2_^−^ contents by 45.2% (MLT vs. MLT + cPTIO in Figure 6B). The results indicated ROS and NO could act antagonistically during the MLT treatment in the mycelium culture under a light/dark shift (24 h:24 h).

### 3.4. Effect of MLT on HA Contents and Gene Expression for HA Biosynthesis

Exogenous MLT significantly reduced HA contents in mycelium culture under a light/dark shift (24 h:24 h) (MLT vs. control in Figure 7A). However, the exogenous applied H_2_O_2_ or NO scavenger cPTIO caused enhancement of HA contents during the MLT treatment (MLT vs. MLT + H_2_O_2_ or MLT + cPTIO in Figure 7B). HA contents were suppressed further by Vc or NO donor SNP (MLT vs. MLT + Vc or MLT + SNP in Figure 7B). The expression levels of seven key HA biosynthesis genes, including polyketide synthase (*PKS*), FAD/FMN-containing dehydrogenase (*FAD*), multicopper oxidase (*MCO*), major facilitator superfamily (*MFS*), *O*-methyl-transferase (*Omef*), zinc finger transcription factor (*ZFTF*) and monooxygenase (*Mono*) were quantitatively detected by qRT-PCR. They were significantly down-regulated by MLT treatment, about 2.57-, 1.93-, 3.74-, 1.97-, 3.57-, 9.73- and 5.30-fold of the control group, respectively (Figure 7B). After being pre-treated with cPTIO and H_2_O_2_ for 30 min prior to MLT treatment, the expression levels of some genes (*MCO*, *Mono*, *ZFTF* and *Omef*) were partially rescued. These results indicated the mediation of ROS and NO in the regulation of MLT on *Shiraia* HA biosynthesis.

## 4. Discussion

MLT, a neuro-hormone in vertebrates, exhibits many hormone-like activities in plants, including regulating seed germination, root growth, circadian rhythm, and fruit ripening, and increasing tolerance to a wide range of adverse environmental factors such as UV irradiation, heavy metal pollutants, extreme temperatures, drought and microbial pathogens [44]. MLT plays a crucial role in protecting plants from oxidative damage in such stresses by improving the efficiency of the mitochondrial electron transport chain, scavenging free radicals, and increasing antioxidant enzyme activities [10]. In plants, the biosynthesis of secondary metabolites can be considered a defense strategy to cope with stressful conditions. Some secondary metabolites, such as volatile terpenoids, phenolics, anthocyanins and flavonoids, can scavenge free radicals and act as nonenzymatic antioxidants against oxidative stress [45]. Recent studies demonstrated that MLT can promote the biosynthesis of such secondary metabolites under stress. The contents of flavonoids and the total phenolic compounds could be enhanced by exogenous MLT applied on apple (*Malus hupehensis*) after UV-B exposure [46], *Hyoscyamus pusillus* callus and citrus under drought [47,48], and garden thyme (*Thymus daenensis*) leaves under salinity stress [49]. Higher phenol, anthocyanin, and flavonoid contents were induced by MLT at 50 μM in rosemary seedlings under arsenic stress [50]. However, in this study, the production of a fungal PQ, HA in *Shiraia* sp. S9 was inhibited significantly by MLT. Hypocrellins are fungal PQ derivatives with no antioxidant activity but with photosensitive activity to produce ROS [24]. The first step of the biosynthesis pathway of HA is the condensation and decarboxylation of acetyl-CoA and malonyl-CoA to nor-toralactone by PKS [42]. Mono catalyzes *O*-methylation, ring-opening, decarboxylation, and hydroxylation reactions formation, and then Omef may likely catalyze methyl groups into the HA backbone. We further confirmed that the transcriptional level of genes (*PKS*, *Mono*, *ZFTF*, *FAD*, *Omef*, *MCO* and *MFS*) in the putative gene clusters for HA biosynthesis were all downregulated significantly in S9 cultures after MLT treatment, resulting in the inhibition of HA contents (Figure 7). Our results provided a new mode for the indirect antioxidant action of MLT by decreasing the reactant (hypocrellin) of the photooxidative reaction for ROS generation. In addition, MLT at 100.00 μM inhibited conidiation and spore germination of *Shiraia* sp. S9 (Figure 1C,D). This is in agreement with the finding of the inhibition of the formation and germination of conidial spores in *Fusarium graminearum* by MLT and its chemical homolog 5-methoxyindole [51]. To the best of our knowledge, our study is the first to assess the effect of MLT on fungal secondary metabolism.

In our previous study [34], the light/dark shift induced the generation of ROS by up-regulating the expression levels of ROS-related genes of NADPH oxidase (*NOX*) and cytochrome c peroxidase (*CCP*). The elevated ROS generation was also confirmed in the green fluorescence of DCFH-DA stained hyphae (Figure 3A). Although MLT treatment did not change the activity of NOX, which catalyzes the NADPH-dependent reduction of molecular oxygen to form superoxide radical anions (Figure 4A), the supplementation of MLT significantly reduced the ROS (H_2_O_2_ and O_2_^−^) levels in the culture under the light/dark shift (Figure 3C,D). In this study, MLT treatment induced an increase in the activities of SOD, CAT, and POD at the different stages of the mycelium culture (Figure 4B–D). As enzyme SOD could efficiently catalyze the transformation of superoxide to H_2_O_2_, which in turn is detoxified into water and oxygen by CAT and POD [52], the enhanced activities of those antioxidant enzymes were responsible for attenuating ROS levels by MLT in the mycelia (Figure 3). Simultaneously, our study has shown the MLT-induced NO generation during the culture (Figure 5A–C). The induced NO production in *Shiraia* sp. S9 was inhibited by the NOS inhibitor L-NAME and NR inhibitor STD, suggesting the involvement of an NOS-like enzyme and NR-dependent side reaction for fungal NO generation [53]. Recently MLT has also been reported to function cooperatively with NO to regulate plant growth, development and defense responses [54]. The generation of NO induced by exogenous MLT was reported to enhance cold tolerance in tomato fruits [55] and to improve tolerance to lead toxicity in maize [56]. Exogenously applied MLT increased NO levels of pepper (*Capsicum annuum*) and alleviated the detrimental effects of salt stress and iron deficiency [57]. NO can act as a downstream signal of MLT in plants and mitigate harmful effects of different stresses, including salinity, heavy metals, drought and osmotic stress [54]. However, there is no information in the literature on the effect of MLT-induced NO on fungal growth and metabolism. Our study is also the first report on the induced generation of NO signal in fungi by MLT. The suppression of MLT-induced O_2_^−^ production by NO donor SNP (MLT + SNP vs. MLT in Figure 6B) and the increase by NO scavenger cPTIO (MLT + cPTIO vs. MLT in Figure 6B) in our study suggests an antioxidant effect of intracellular NO. In addition, the scavenger ROS by Vc could lead to enhanced NO levels during MLT treatment (MLT + Vc vs. MLT in Figure 6A). These results suggested a possible antagonistic action between NO and ROS during MLT treatment. Deng et al. (2016) reported that exogenous H_2_O_2_ at 10–20 mM also played positive effects on HA production of *S. bambusicola* [28]. Recently, we demonstrated that endogenous ROS generation was one of the early signals for the elicitation of HA production in *Shiraia* mycelium cultures under the ultrasound treatment [33], a light/dark shift [34] and lanthanum (La^3+^) [58]. Therefore, the existence of interplays between NO and ROS might have been involved in HA biosynthesis during MLT treatment. Our results showed that H_2_O_2_ or NO scavenger cPTIO could rescue MLT-induced inhibition on expressions of HA biosynthetic genes and their content (Figure 7). These findings further support the involvement of NO and ROS in HA biosynthesis during MLT treatment. In the proposed hypothetical model for MLT regulation (Figure 8), MLT, antioxidant enzymes and NO cooperate to enhance their antioxidative potentials in the cultures of *Shiraia* sp. S9 under a 24 h:24 h light-dark shift, leading to the suppression of ROS signals for the elicitation of HA biosynthesis and the final inhibition of HA production.

ROS generation can be triggered by HA under irradiation and exert oxidative stress on HA-yielding fungus of *S. bambusicola* [28]. However, *Shiraia* with higher HA production can retain growth and morphology in normal and even high biomasses under visible light irradiation [34,59], indicating an excellent antioxidant system of the fungus. Vitamin B6 (pyridoxine) biosynthesis was proved to be vital for the resistance of *Cercospora nicotianae* to its own abundant ROS produced by its cercosporin, a hypocrellin-like PQ compound [60]. Antioxidants (cysteine, ascorbate and reduced glutathione) and antioxidant enzymes also enhanced the resistance of *C. nicotianae* to the phototoxicity of cercosporin [61]. Callahan et al. (1999) reported that a cercosporin facilitator protein (CFP) contributed self-resistance to self-produced cercosporin by actively exporting the toxin out of cells [62]. Our results suggested a new possible mechanism for *Shiraia* self-resistance via MLT biosynthesis to inhibit phototoxic HA compounds. The changes of endogenous MLT contents in *Shiraia* at different growth stages and to different environmental factors need further investigation. On the other hand, PQ phototoxins such as HA, cercosporin or elsinochrome are a pathogenicity factor for fungal infection via ROS generation, damaging the cells of host plants [25]. Photoactivated PQs are biosynthesized overwhelmingly by plant parasitic fungi, including *Shiraia*, *Cercospora* and *Elsinoё* species, which cause economically important diseases on many important plants such as citrus, corn, and soybean as well as vegetable crops and horticultural plants [63]. Thus, it is a plausible strategy to combat these fungal diseases by suppressing the production of photoactive PQ. *Shiraia* spp. are pathogenic fungi parasitizing mainly on more than 10 species of bamboos [64]. In the present study, our results showed the strong adverse activity of MLT against *Shiraia* sp. S9 by inhibiting HA production (Figure 1 and Figure 2). Many previous investigations have demonstrated the anti-pathogenic activity of MLT in animals, e.g., reducing the deleterious effects of the Venezuelan equine encephalomyelitis virus, decreasing blood and brain viruses in mice and enhancing bactericidal capacity against multidrug-resistant bacteria [65]. MLT was also reported to increase fungicide susceptibility and enhance the vulnerability of phytopathogenic fungi (*Botrytis*, *Penicillium*, *Fusarium*, *Phytophthora* and *Alternaria* spp.) [66]. Therefore, the combined application of MLT and other fungicides could be developed as a novel biocontrol strategy to control plant diseases caused by PQ-producing fungi.

## 5. Conclusions

In summary, this study presented the first assessment of the inhibitive action of exogenous MLT on *Shiraia* conidiation, germination and HA biosynthesis. Although MLT has recently been considered a novel regulator (hormone) of plant growth and development and an important antioxidant against oxidative stress, the regulation of MLT on fungal growth and metabolites has not been well studied. Our study clearly showed MLT could enhance the activities of antioxidant enzymes and reduce the increased generation of ROS in the mycelium culture of *Shiraia* sp. S9 under a light/dark shift (24 h:24 h). It is an interesting finding that MLT induced endogenous NO generation in the hyphae via NOS- or NR-dependent routes. MLT-induced NO, a downstream signaling molecule, was implicated in the melatonin-promoted antioxidant responses in the mycelia of *Shiraia* sp. S9. NO and ROS mediated MLT-induced downregulation of the transcript levels of HA biosynthetic genes, resulting in the inhibition of HA production. Our study provided a new understanding of self-resistance in phototoxin-producing fungi. These findings could be harnessed for the potential agricultural application of MLT to control plant diseases caused by PQ-producing fungi.

## Figures and Tables

**Figure 1 jof-08-00836-f001:**
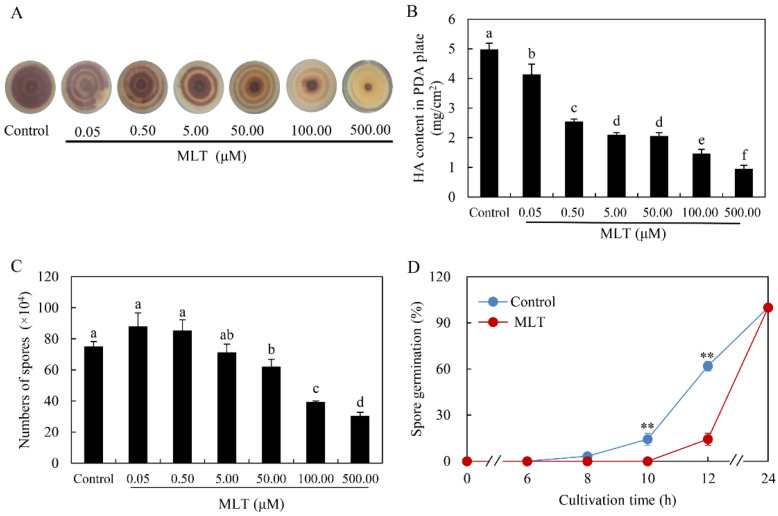
The effects of melatonin (MLT) on growth and HA production of *Shiraia* sp. S9 in solid culture. (**A**) Fungal colony in the PDA plates with MLT at 0.05–500.00 μM. The culture was maintained at 28 °C for 8 days under light/dark shift (24 h:24 h). HA content (**B**) and the number of spores (**C**) in the PDA plate after MLT treatment. Values are mean ± SD from three independent experiments. Different letters above the bars mean significant differences (*p* < 0.05). (**D**) Effect of MLT at 100 μM on fungal spore germination. Values are mean ± SD from three independent experiments (** *p* < 0.01 vs. control).

**Figure 2 jof-08-00836-f002:**
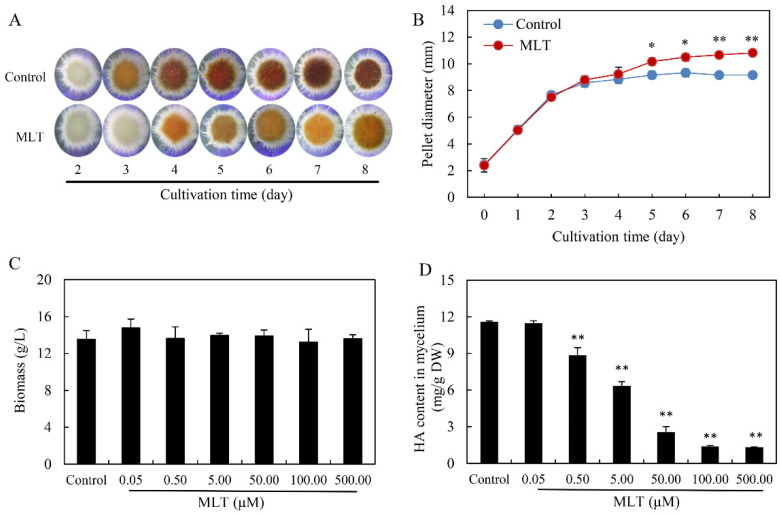
The effects of melatonin (MLT) on growth and HA production of *Shiraia* sp. S9 in liquid culture. (**A**) The pellet morphology (15×) and pellet diameters (**B**) were measured during the culture. MLT at 100.00 μM was added on day 2 of the culture at 28 °C for 8 days under a light/dark shift (24 h:24 h). Effect of different MLT concentrations on fungal biomass (**C**) and HA production (**D**) in the culture. Values are mean ± SD from three independent experiments (* *p* < 0.05 and ** *p* < 0.01 vs. control).

**Figure 3 jof-08-00836-f003:**
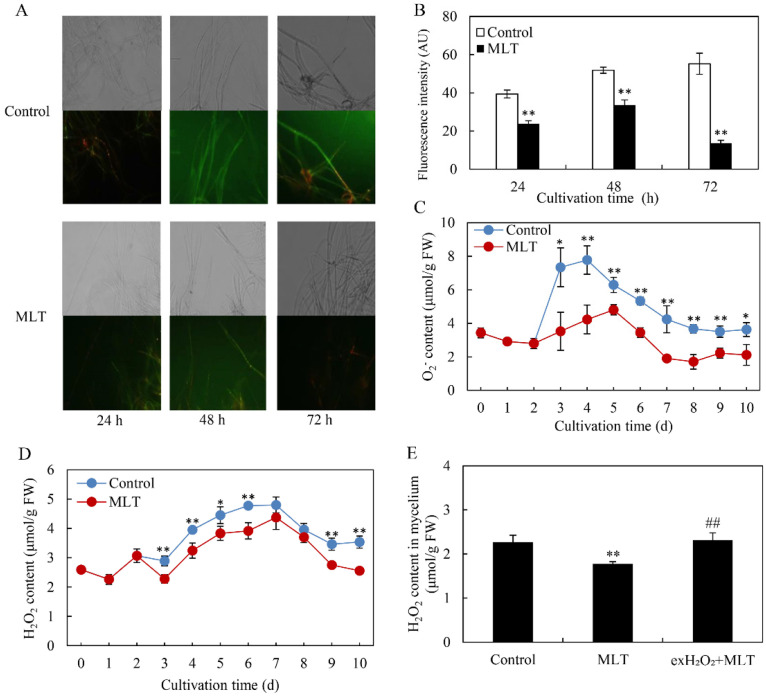
The effects of melatonin (MLT) on ROS production of *Shiraia* sp. S9 in liquid culture. (**A**) Bright-field image (above) and fluorescence microscopy of DCFH-DA-stained mycelia (below) (400×) in the cultures. The culture was maintained in a 150 mL flask containing 50 mL medium at 150 r/min and 28 °C under light/dark shift (24 h:24 h). MLT at 100 μM was added on day 2 of the culture. (**B**) Fluorescence intensity of ROS stained by DCFH-DA after 100 μM MLT treatment. Time course of O_2_^−^ (**C**) and H_2_O_2_ contents (**D**) in S9 mycelia. (**E**) H_2_O_2_ content in mycelia after MLT treatment with exogenous H_2_O_2_ for 24 h. H_2_O_2_ (100 μM) was added 30 min prior to MLT treatment. Values are mean ± SD from three independent experiments (* *p* < 0.05 and ** *p* < 0.01 vs. control, ^##^
*p* < 0.01 vs. MLT treatment).

**Figure 4 jof-08-00836-f004:**
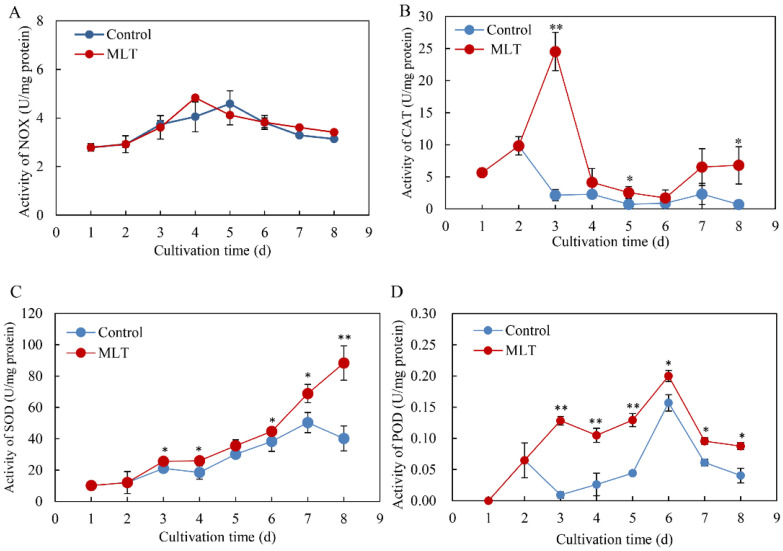
Effects of melatonin (MLT) on enzyme activities of NADPH oxidase (NOX) (**A**), catalase (CAT) (**B**), superoxide dismutase (SOD) (**C**), and peroxidase (POD) (**D**) in the mycelia of *Shiraia* sp. S9 in liquid culture. The culture was maintained in a 150 mL flask containing 50 mL medium at 150 r/min and 28 °C under a light/dark shift (24 h:24 h) for 8 days. MLT at 100.00 μM was added on day 2 of the culture. Values are mean ± SD from three independent experiments. (* *p* < 0.05 and ** *p* < 0.01 vs. control).

**Figure 5 jof-08-00836-f005:**
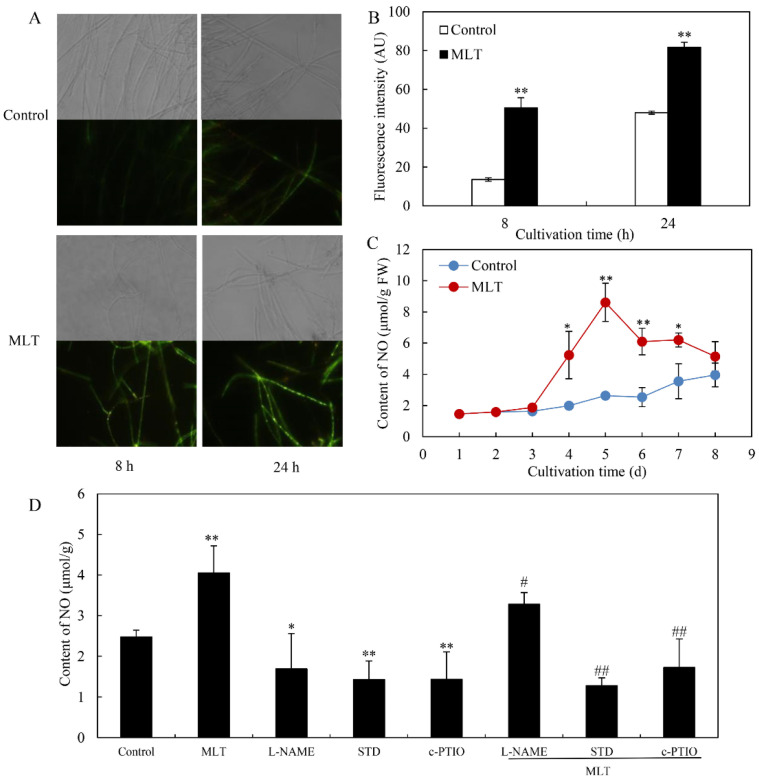
The effect of melatonin (MLT) on NO generation of *Shiraia* sp. S9 in liquid culture. (**A**) Bright-field image (above) and fluorescence microscopy of DAF-2 DA-stained mycelia (below) (400×) in the cultures. The mycelia were treated with MLT (100.00 μM) on day 2 of the culture under a light/dark shift (24 h: 24 h) at 28 °C. The relative fluorescence intensity (**B**) and NO contents (**C**) in mycelia. (**D**) Effect of NO inhibitors on MLT-induced NO production. L-NAME (100 μM), STD (100 μM) and cPTIO (100 μM) were added 30 min prior to MLT treatment, respectively. NO content was detected after 24 h of MLT treatment. Values are mean ± SD from three independent experiments (* *p* < 0.05 and ** *p* < 0.01 vs. control. ^#^
*p* < 0.05 and ^##^
*p <* 0.01 vs. MLT treatment).

**Figure 6 jof-08-00836-f006:**
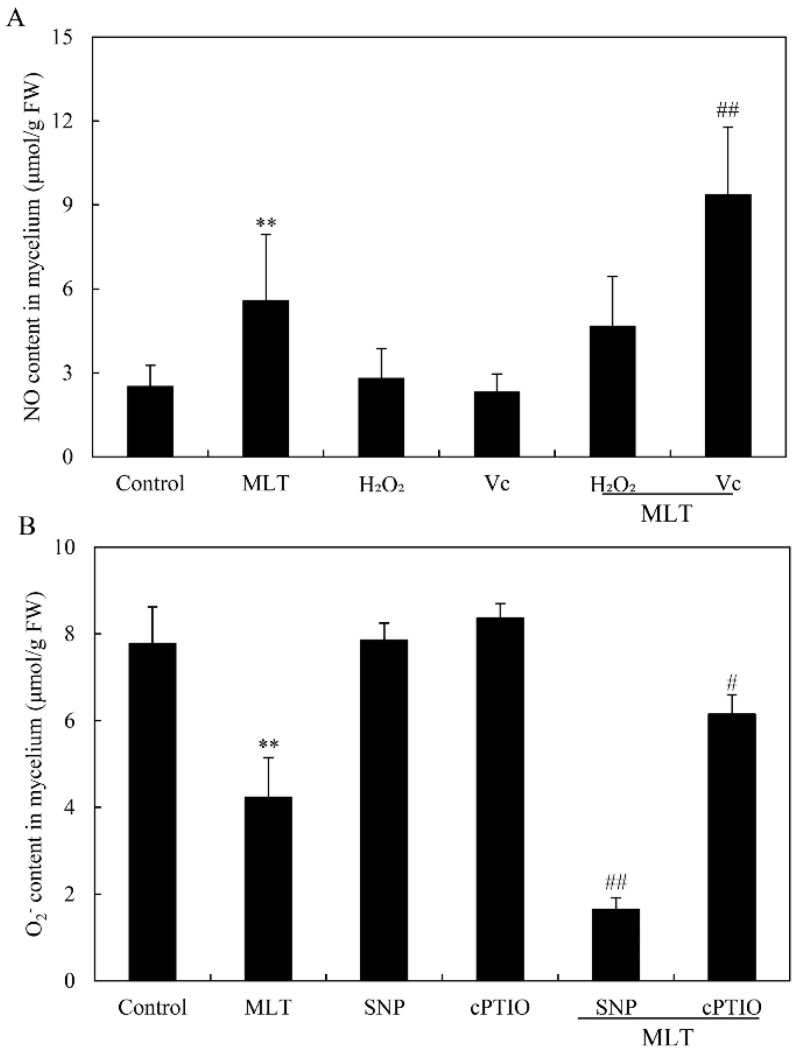
Effect of melatonin (MLT) on NO (**A**) and O_2_^−^ contents (**B**) in the cultures of *Shiraia* sp. S9 and the influence of the donor and scavengers of NO and ROS. The culture was maintained in a 150 mL flask containing 50 mL medium at 150 r/min and 28 °C under a light/dark shift (24 h:24 h). MLT at 100.00 μM was added on day 2. H_2_O_2_ (100 μM), Vc (10 μM), SNP (10 μM) and cPTIO (100 μM) were added 30 min prior to MLT, and the contents were detected after 24 h of MLT treatment. Values are mean ± SD from three independent experiments (** *p* < 0.01 vs. control. ^#^
*p <* 0.05 and ^##^
*p* < 0.01 vs. MLT treatment).

**Figure 7 jof-08-00836-f007:**
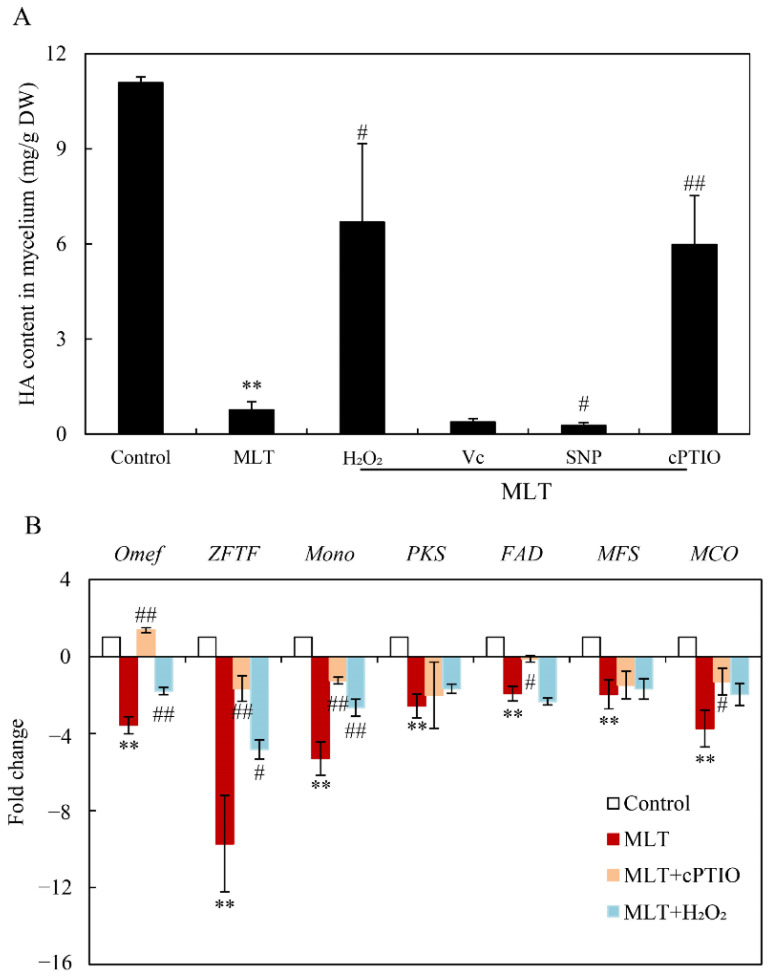
Effects of melatonin (MLT) on HA production (**A**) and the gene expressions (**B**) for HA biosynthesis genes of *Shiraia* sp. S9. The culture was maintained in a 150 mL flask containing 50 mL medium at 150 r/min and 28 °C under a light/dark shift (24 h: 24h) and harvested on day 8. MLT (100.00 μM) was added on day 2 of the culture. H_2_O_2_ (100 μM), Vc (10 μM), SNP (10 μM) and cPTIO (100 μM) were added to the culture 30 min prior to MLT treatment. Values are mean ± SD from three independent experiments (** *p* < 0.01 vs. control, ^#^
*p <* 0.05 and ^##^
*p <* 0.01 vs. MLT treatment).

**Figure 8 jof-08-00836-f008:**
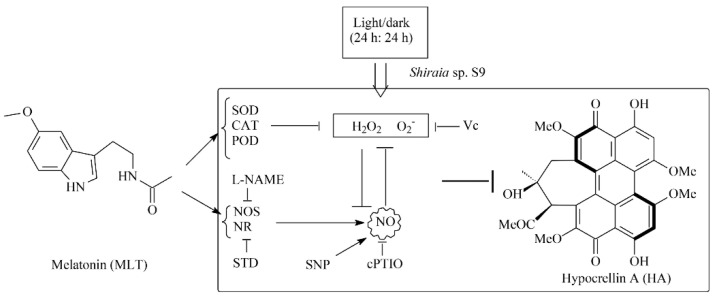
A simplified model for the inhibition of hyprocrellin A (HA) by melatonin (MLT). Black lines with arrows indicate promotion of pathway or expression, and lines with blocked ends indicate inhibition of pathway or expression.

## Data Availability

Not applicable.

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
