# Peer review of "Melatonin-Induced Inhibition of Shiraia Hypocrellin A Biosynthesis Is Mediated by Hydrogen Peroxide and Nitric Oxide"

_jof, 2022, doi:10.3390/jof8080836_

Round 1

Reviewer 1 Report

Manuscript of Wang et all. presents results concerning the role of exogenous melatonin on growth, development and oxidative  stress response in pathogenic fungus Shiraia bambusicola.  The subject is generally interesting, as the role of melatonin in fungi is not well known. Authors showed that melatonin enhances activities of antioxidative enzymes and, interestingly, induced endogenous NO generation. By regulating the ROS and NO levels, melatonin indirectly downregulates transcription of hypocrellin A biosynthetic genes. Authors propose an interesting mechanism of self-resistance in phototoxin-producing fungi.

In my opinion, before publication, few things should be corrected.

1. The major point relates to the figure 3 which evidently lacks part D and E :

l. 217-218- " ... and H2O2 contents (D) in 217 S9 mycelia. (E) H2O2 content in mycelia after MLT treatment with exogenous H2O2 for 24 218 h". Moreover these not shown data are further described in Results (l.206-208) and discussed in Discussion (l.348-349).

2. In Introduction, biosynthesis of hypocrellin A should be briefly described with respective references, taking into account the role of seven genes, which expression is tested in part 3.4 and respective gene clusters.

Minor points:

l. 282 - it would better - " exogenous melatonin significantly reduced the HA content in mycelim..."

l. 287 - it would better - "The expression levels of seven key HA biosynthesis genes, including plolyketide synthase...."

l.293-294 - correct - the expression levels of some genes (MCO, Mono, 293 ZFTF and Omef) were partially rescued.

Author Response

RESPONSE: First of all, I would like to express my sincere thanks for your review. Your review contributes the great improvement of our manuscript. Thank you very much again for paying your valuable time for reviewing our manuscript. The correction and all changes (highlighted in yellow color in MS) were performed according to your comments and showed as follows:

Reviewer(s)' Comments to Author:

Reviewer: 1

Comments and Suggestions for Authors

Manuscript of Wang et all. presents results concerning the role of exogenous melatonin on growth, development and oxidative stress response in pathogenic fungus Shiraia bambusicola. The subject is generally interesting, as the role of melatonin in fungi is not well known. Authors showed that melatonin enhances activities of antioxidative enzymes and, interestingly, induced endogenous NO generation. By regulating the ROS and NO levels, melatonin indirectly downregulates transcription of hypocrellin A biosynthetic genes. Authors propose an interesting mechanism of self-resistance in phototoxin-producing fungi.

In my opinion, before publication, few things should be corrected.

Q1 The major point relates to the figure 3 which evidently lacks part D and E:

217-218- " ... and H2O2 contents (D) in S9 mycelia. (E) H2O2 content in mycelia after MLT treatment with exogenous H2O2 for 24 h". Moreover these not shown data are further described in Results (l.206-208) and discussed in Discussion (l.348-349).

--RESPONSE: Sorry for the missing Fig. 3D and 3E. It happened when we attached the figures to the journal template. We added the correct figure.

Q2 In Introduction, biosynthesis of hypocrellin A should be briefly described with respective references, taking into account the role of seven genes, which expression is tested in part 3.4 and respective gene clusters.

--RESPONSE: We added a description on key genes for HA biosynthesis in Discussion section (line 347-351).

Q3 282 - it would better - " exogenous melatonin significantly reduced the HA content in mycelim..."

--RESPONSE: We rephrased the sentence (line 297-298). 

Q4 287 - it would better - "The expression levels of seven key HA biosynthesis genes, including plolyketide synthase...."

--RESPONSE: We rephrased the sentence (line 302).

Q5 293-294 - correct - the expression levels of some genes (MCO, Mono, ZFTF and Omef) were partially rescued.

--RESPONSE: We rephrased the sentence (line 309).

Reviewer 2 Report

The authors present a study on the effects of melatonin, a known animal and plant hormone, but scarcely studied in fungi, on the development and toxin production of a bamboo phytopathogen. The biochemical relevant pathways were also explored.

This makes a sound and novel contribution to the field.

However, there are some issues that must be attended before accepting the paper:

Lines 45-46: It is understandable that NO could induce anti-oxidant enzymes and protect the plant against pathogens (causing an oxidative stress to the pathogen). However NO is a Nitrogen reactive species by itself, How could this molecule alleviate oxidative stress? (for example in salt-stressed plants). It is known in plants that NO at certain concentrations can act as a signal molecule without causing oxidative damage, maybe a phrase describing these role for NO would make the sentence less confusing.

Line 85: A brief description of the medium would be friendlier to the the reader, otherwise it has to go the the cited article (which is not an Open Access publication). Is it a minimal mineral medium? Is it a rich medium? Just this information may suffice. A more curious reader can go to the cited paper for details and media preparation

Line 90: Again, this is not an Open Access publication, a short description would be empathic with the reader, specially considering that there is not too much work regarding melatonin in microorganisms

Line 168: It seems that HA content does not strictly decreases  in a a dose dependent-manner, at least for some concentrations.  For example, the 5.00 and 50.00 look much alike, and the difference from 50 to 100. Is it really half the value? It looks higher...Maybe the authors could also test if there are statistical significant differences between MLT treatments (not only against the control value) or plot their data as a curve (instead of bars), in which the equation should show linearity (with a high R value) if it is dose dependent

Line 170: Does this suggests that many of the 90% germinated conidias at 24 h in panel D are unable to progress through development and can not reach the sporulation phase? If it is the case, this finding would be worth to highlight

Line 171: In this regard also, MLT did not impair sporulation, since at 24 hours 90% is achieved in both conditions

Line 207: Panels D and E are missing in Figure 3 ! , please add them...

Line 252: correct Typo, the numbers in "H2O2" should be underscript

Line 332: are these molecules necessary for bamboo pathogenesis? If so, please include a sentence in the Introduction

Author Response

RESPONSE: First of all, I would like to express my sincere thanks for your review. Your review contributes the great improvement of our manuscript. Thank you very much again for paying your valuable time for reviewing our manuscript. The correction and all changes (highlighted in yellow color in MS) were performed according to your comments and showed as follows:

Reviewer(s)' Comments to Author:

Reviewer 2

Comments and Suggestions for Authors

The authors present a study on the effects of melatonin, a known animal and plant hormone, but scarcely studied in fungi, on the development and toxin production of a bamboo phytopathogen. The biochemical relevant pathways were also explored.

This makes a sound and novel contribution to the field.

However, there are some issues that must be attended before accepting the paper:

Q1 Lines 45-46: It is understandable that NO could induce anti-oxidant enzymes and protect the plant against pathogens (causing an oxidative stress to the pathogen). However NO is a Nitrogen reactive species by itself, How could this molecule alleviate oxidative stress? (for example in salt-stressed plants). It is known in plants that NO at certain concentrations can act as a signal molecule without causing oxidative damage, maybe a phrase describing these role for NO would make the sentence less confusing.

--RESPONSE: We added a phrase describing dual functions (toxic or protective) of NO in plants under the stresses in Introduction section (line 41-47).

Q2 Line 85: A brief description of the medium would be friendlier to the reader, otherwise it has to go the cited article (which is not an Open Access publication). Is it a minimal mineral medium? Is it a rich medium? Just this information may suffice. A more curious reader can go to the cited paper for details and media preparation.

--RESPONSE: We added medium composition in the section of Materials and Methods (line 94-95).

Q3 Line 90: Again, this is not an Open Access publication, a short description would be empathic with the reader, specially considering that there is not too much work regarding melatonin in microorganisms.

--RESPONSE: We presented a brief introduction on light condition in the section of Materials and Methods (line 98-101).

Q4 Line 168: It seems that HA content does not strictly decreases in a dose dependent-manner, at least for some concentrations. For example, the 5.00 and 50.00 look much alike, and the difference from 50 to 100. Is it really half the value? It looks higher...Maybe the authors could also test if there are statistical significant differences between MLT treatments (not only against the control value) or plot their data as a curve (instead of bars), in which the equation should show linearity (with a high R value) if it is dose dependent.

--RESPONSE: The effect of MLT on HA contents is not in a concentration dependent manner in Fig. 1B. We presented the statistical significance among the control and treatment groups with the different letters above the bars in Fig. 1B and 1C.

Q5 Line 170: Does this suggests that many of the 90% germinated conidia at 24 h in panel D are unable to progress through development and can not reach the sporulation phase? If it is the case, this finding would be worth to highlight. In this regard also, MLT did not impair sporulation, since at 24 hours 90% is achieved in both conditions.

--RESPONSE: They are different experiments for spore germination (Fig. 1D) and the amount of conidiation production from the mycelia (Fig. 1C). In Fig. 1C the fungal mycelia were treated by MLT at 100 μM for 8 days, and the conidia were quantified. In Fig. 1D, the collected spores were treated by MLT at 100 μM and observed on the germination. Current results can’t support the speculation that “90% germinated conidia at 24 h in panel D are unable to progress through development and can not reach the sporulation phase”. Although spores from control and MLT group can achieve a germination rate 90% after 24 h, the significant inhibition also observed during 10-12 h.

Q6 Line 207: Panels D and E are missing in Figure 3 ! , please add them...

--RESPONSE: Sorry for the missing Fig. 3D and 3E. It happened when we attached the figures to the journal template. We added the correct figure.

Q7 Line 252: correct Typo, the numbers in "H2O2" should be underscript.

--RESPONSE: We correct (line 268).

Q8 Line 332: are these molecules necessary for bamboo pathogenesis? If so, please include a sentence in the Introduction.

--RESPONSE: We included a sentence and cited reference [25] for phototoxic hypocrellin as virulence factor of bamboo pathogenesis (Daub et al. (2013). Reactive oxygen species in plant pathogenesis: the role of perylenequinone photosensitizers. Antioxidants & Redox Signaling, 19(9), 970-989.) in Introduction section (line 71-73).

Reviewer 3 Report

See the attachment please

Author Response

RESPONSE: First of all, I would like to express my sincere thanks for your review. Your review contributes the great improvement of our manuscript. Thank you very much again for paying your valuable time for reviewing our manuscript. The correction and all changes (highlighted in yellow color in MS) were performed according to your comments and showed as follows:

Reviewer(s)' Comments to Author:

Reviewer: 3

Q1 Possible contradiction is that "MLT treatment was associated with higher activity of antioxidant enzymes (superoxide dismutase, catalase and peroxidase) and marked decline of reactive oxygen species (ROS) production in the mycelia" (Line 16) and " H2O2 was proved to be involved in the MLT-induced downregulation of the expressions of HA biosynthetic genes, leading to the suppression of HA production " (Line 22). But higher levels of both catalase and peroxidase (Figure 4) mean decomposition or consumption of H2O2 in the biosystem. Since that the hydrogen peroxide active participation in the suppression of HA production remains questionable. Moreover, "the exogenous applied H2O2 caused enhancement of HA contents during the MLT treatment" (Line 283). Please provide the explanatory context for these controversial phrases.

--RESPONSE: Maybe some sentences in our MS make the confusion. We presented our research thought here. The previous studies demonstrated the exogenous applied H2O2 or induced oxidative stress can enhance fungal HA. In this study, light/dark shift (24 h: 24 h) caused ROS generation and HA accumulation. But exogenous MLT treatment could increase activities of antioxidative enzymes and induce NO generation to make antagonistic effect against ROS generation, leading to the inhibition of HA production. We rephrased the sentence as “The changes of NO and H2O2 were proved to…” in Abstract (Line 22-23).

Q2 Line 22: "The NO donor sodium nitroprusside (SNP) potentiated MLT-induced inhibition of [superoxide radical anion] production, but NO scavenger cPTIO enhanced it, whereas ...". This sentence sounds controversially, and should be better rewritten as "The NO donor sodium nitroprusside (SNP) potentiated MLT-induced inhibition of [superoxide radical anion] production, but NO scavenger cPTIO enhanced [superoxide radical anion] production, whereas ..."

--RESPONSE: We rephrased the sentence according to the reviewer’s suggestion (line 20-22).

Q3 Line 97: " MLT (0.05-500.00 μM) was added to the liquid medium on day 2 of the culture". However, Figures 2 (B), 3 (C) contain a red line "MLT" starting from "0 day" of cultivation time at abscissa axis; Figures 4 (A-D), 5 (C) contain a red line "MLT" starting from "1 day" of cultivation time at abscissa axis. Moreover, that is not due to a subtraction of "period without MLT" from the corresponding curves, since in the Figure 4 caption we see "The culture was maintained ... for 8 days.", and namely day 8 is depicted as the last day of cultivation time (Figure 4 (A-D), Figure 5 (C)).

--RESPONSE: To observe the changes during the culture, we included “0 day” of the initial culture and “1 day” before the MLT addition. MLT was added to the liquid medium on day 2 of the culture. We included the explanation in the figure legend.

Q4 Line 138: "donor and scavenger" should be replaced by "donor and scavenger, respectively".

--RESPONSE: We rephrased the sentence (line 151). 

Q5 Figure 2 (B), ordinate axis: "Diameter" should be replaced by "Pellet diameter".

--RESPONSE: We added “Pellet diameter” in ordinate axis of figure 2B. 

Q6 In Figure 2, hypocrellin A content in liquid-culture mycelium is given in mg/g of DRY mass; in the Figure 3 (C), content of [superoxide radical anion] is given in micromole/g of FRESH mass; in the Figure 5 (C), NO content is given in micromole/g of FRESH mass. Therefore, the biomass treatment and the application of mycelia with varying water content for analyses should be substantiated in Materials and Methods.

--RESPONSE: We used dry mass for the HA content, and fresh mess for the NO content and some enzyme activity. That is used according the different unit definition. We think it is suitable to using the same unit for the comparison between the control and treated group. Certainly the reviewer gives a full consideration on the effect of MLT on water contents in fungal mycelia. The study on the MLT-induced changes of primary metabolites needs further investigation in future.

Q7 Figures 3 (B), 5 (B): "Mean fluorescence intensity" axis – what are units? If arbitrary units, please indicate as "AU" in the Figure, and mention "fluorescence arbitrary units" once in Materials and Methods in any form.

--RESPONSE: We added the AU as the unit in these Figures.

Q8 Figure 3 (D): H2O2 contents in S9 mycelia is not presented, but listed in Figure caption. Figure 3 (E): H2O2 content in mycelia after MLT treatment with exogenous H2O2 for 24 h is not presented, but listed in Figure caption. It is especially unfortunate, since lacking parts of Figure 3 could potentially contribute to discussion on point 1 of this review

--RESPONSE: Sorry for the missing Fig. 3D and 3E. It happened when we attached the figures to the journal template. We added the correct figure.  

Round 2

Reviewer 2 Report

The manuscript can be accepted in its present form. Congratulations

Reviewer 3 Report

I believe the manuscript has been significantly improved:

- by rephrasing several sentences, the authors have made the essence of the work much more clear, e.g. "The previous studies demonstrated the exogenous applied H2O2 or induced oxidative stress can enhance fungal HA. In this study, light/dark shift (24 h: 24 h) caused ROS generation and HA accumulation. But exogenous MLT treatment could increase activities of antioxidative enzymes and induce NO generation to make antagonistic effect against ROS generation, leading to the inhibition of HA production.";

- the authors have included the explanation in the Figures 4 and 5 legends related to the observation period of culture growth;

- the authors have added the correct Figure 3 with all its parts.